# Expanding the E-Liquid Flavor Wheel: Classification of Emerging E-Liquid Flavors in Online Vape Shops

**DOI:** 10.3390/ijerph192113953

**Published:** 2022-10-27

**Authors:** Shaoying Ma, Zefeng Qiu, Qian Yang, John F. P. Bridges, Jian Chen, Ce Shang

**Affiliations:** 1Center for Tobacco Research, The Ohio State University Wexner Medical Center, Columbus, OH 43214, USA; 2Department of Computer Science and Engineering, The Ohio State University, Columbus, OH 43210, USA; 3Department of Biomedical Informatics, The Ohio State University Wexner Medical Center, Columbus, OH 43210, USA; 4Department of Internal Medicine, The Ohio State University Wexner Medical Center, Columbus, OH 43210, USA

**Keywords:** vaping, e-liquid, e-cigarette, flavor, electronic nicotine delivery systems, ENDS, electronic cigarette, tobacco, regulation, FDA

## Abstract

Introduction: Electronic cigarettes are the most popular tobacco product among U.S. youth, and over 80% of current youth users of e-cigarettes use flavored e-cigarettes, with fruit, mint/menthol, and candy/sweets being the most popular flavors. A number of new e-liquid flavors are currently emerging in the online e-cigarette market. Menthol and other flavored e-cigarettes could incentivize combustible tobacco smokers to transition to e-cigarette use. Methods: From February to May 2021, we scraped data of over 14,000 e-liquid products, including detailed descriptions of their flavors, from five national online vape shops. Building upon the existing e-liquid flavor wheel, we expanded the semantic databases (i.e., key terms) to identify flavors using WordNet—a major database for keyword matching and group discussion. Using the enriched databases, we classified 14,000+ e-liquid products into the following 11 main flavor categories: “fruit”, “dessert/candy/sweets”, “coffee/tea”, “alcohol”, “other beverages”, “tobacco”, “mint/menthol”, “nuts”, “spices/pepper”, “other flavors”, and “unspecified flavor”. Results: We find that the most prominent flavor sold in the five online vape shop in 2021 was fruit flavored products, followed by dessert/candy/other sweets. Online vendors often label a product with several flavor profiles, such as fruit and menthol. Conclusions: Given that online stores market products with multiple flavor profiles and most of their products contain fruit flavor, the FDA may have issued marketing denial orders to some of these products. It is important to further examine how online stores respond to the FDA flavor restrictions (e.g., compliance or non-compliance).

## 1. Introduction

Electronic cigarettes are a group of novel nicotine or tobacco products that have rapidly gained popularity in recent years, especially among US adolescents and young adults [1,2]. An e-cigarette is a device that evaporates a liquid solution called e-liquid, so that the user could inhale vapor [3,4]. Flavor is one of the key attributes of e-liquids, especially to youth users of e-cigarettes [5,6]. Menthol flavor in nicotine e-liquids may counter the aversiveness of nicotine through its cooling effect, and for nicotine-free e-liquid products, fruit flavor may increase the product appeal to cigarette smokers [7,8]. Adolescents are interested in trying e-cigarettes with fruit flavor compared to tobacco or alcohol flavor, partially due to the perception that fruit-flavored e-cigarettes are less harmful than tobacco-flavored e-cigarettes [9]. Despite flavors being one of the main reasons why youth initiate e-cigarette use, they are often not aware of the nicotine level in the e-cigarettes they use [10]. Development of new e-liquid flavors could be associated with initiation and escalation of e-cigarette use among USA adolescents and adults [5,6,11,12,13]. In total, 82.9% of current youth users of e-cigarettes in the USA used flavored e-cigarettes in 2020, and the most popular flavors are fruit, mint/menthol, and dessert/candy/sweets [14]. The market of e-cigarettes has quickly expanded since mid-2000s, and online vape shops are one of the most common outlets for purchases of e-cigarettes [15,16,17].

Unlike combustible cigarettes, e-cigarettes typically come in a variety of product design, such as open versus closed system, which poses challenges to tobacco control policies [4]. Open system e-cigarettes have refillable tanks and are completely reusable, whereas closed system e-cigarettes are either single-use products (disposable) or reloadable with pre-filled cartridges [18]. Open systems allow users to manually refill cartridges with flavored e-liquids that they purchase, thus giving them customized vaping experience, and closed systems are designed to allow little modification of contents by users [4,19].

E-cigarettes may serve as a gateway to cigarette smoking, through increasing the initiation of nicotine use (that would not occur in the absence of e-cigarettes) and e-cigarette users (who become addicted to nicotine) transitioning from e-cigarettes to other tobacco products including cigarettes [16,20,21,22]. In addition, a growing proportion of e-cigarette users report that they initiated use at a young age (14 years old or younger), and 22.5% of high school users are daily vapers, indicating their addiction to e-cigarettes [14,23]. Regulations on e-liquid flavors are under the authority of the USA Food Drug Administration (FDA). In response to the alarming rise of vaping behaviors especially among youth in the USA, a federal regulation was imposed in February 2020 that bans all flavors in cartridge-based e-cigarettes except for tobacco and menthol, but it has left loopholes for manufacturers to sell menthol-flavored cartridge products, as well as flavored e-liquid products that are used in disposable e-cigarettes and open-system e-cigarettes [24,25,26,27]. In addition, the FDA has been issuing marketing denial orders to manufacturers of e-cigarette products [28]. Several states in the USA have restricted the sale of flavored e-cigarette products, including Massachusetts, New Jersey, New York, and Rhode Island, and their flavor bans were implemented in either 2019 or 2020 [29].

In order to examine e-cigarette marketplace and enforce FDA and local flavor restrictions, identifying and categorizing e-cigarette flavors are critical. Although research efforts have been made to document and classify e-cigarette flavors (e.g., flavor wheel), such efforts may fail to capture the rapid evolvement of e-cigarette marketplace, such as the development in flavor profiles. Moreover, all existing studies on flavors focus on the primary flavor profile of a product (e.g., the flavor labeled on a package), which may not reflect secondary flavors or multiple flavors that could be marketed to consumers [30]. Finally, as the FDA bans flavored cartridges and issues market denial orders to flavored e-cigarette products, manufacturers and retailers may market their products using concept flavors, i.e., ambiguous flavor descriptors to evoke sensory or other pleasant vaping experiences (e.g., jazz, ice, etc.). These challenges need to be evaluated in order to better inform flavor restrictions and marketing regulatory policies.

To address some of the research gaps or regulatory challenges identified above, we conduct this study to achieve the following goals: (1) expanding and enriching the existing flavor wheel to capture the diversity of words that are used to describe flavors (i.e., building a semantic database for flavors); (2) use this semantic database to identify and classify flavors of over 14,000 products sold online. Given that both the e-cigarette market and the regulatory environment have been rapidly evolving, our flavor semantic database could be a very useful tool to identify and classify flavors in real time and used by researchers and policymakers to surveille marketplace and check compliance for flavor restrictions. Thus, the findings of this study can be used not only to inform potential policies, but also to shed light on future research directions regarding e-cigarette flavors.

## 2. Materials and Methods

To build the semantic database for flavors, we use key words documented in the following sources: (1) the e-cigarette flavor wheel created by Krüsemann et al. (2019) [30], which defined the main categories and subcategories of flavors (e.g., fruit, sweet, etc.); (2) WordNet [31], a large lexical database of English in which words are interlinked by semantic relations, which listed keywords for each flavor category (fruit, sweet, etc.); (3) Marketing description data from Ma et al. (2022) [32] where we identify words related to flavors; and (4) group discussion where six coauthors discuss the inclusion and exclusion of each word that WordNet provides.

Using the semantic database, we classified flavors for over 14,000 e-liquid products that we collected from give online stores. Given that each store may describe or list product flavors differently, we obtained the flavor information using three different methods: (1) extracting product flavor(s) directly from source code of product webpage, which is presumably the most accurate; (2) identifying product flavor(s) from flavor filter provided by each vape shop; and (3) extracting product flavor(s) from product description box on list page aided by keyword matching. (See Figure A1 in the Appendix A for an example of flavor filters on a store website). For each product, multiple flavor classifications could be assigned if the marketing description or language mentioned key words that fall into different flavor classifications.

## 3. Results

The key terms we used in each main category and subcategory are summarized in Table A1. Figure 1 is a frequency plot of the e-liquid products in our sample based on whether they contain a specific flavor. As shown in both Appendix A Table A1 and Figure 1, we classify e-liquid flavors into 11 main categories, i.e., fruit, dessert/candy/other sweets, tobacco, menthol/mint, nuts, spices/pepper, coffee/tea, alcohol, other beverages, other flavors, and unflavored. Within the main category of fruity flavors, there are four subcategories: berry, citrus, tropical, and other fruits. We present in Figure 1 that, out of the 14,477 e-liquid products in our sample, 12,291 of them contain at least one fruity flavor. There are 7755 e-liquid products from the five online vape shops that contain dessert/candy/other sweet flavor(s). A total of 3043 e-liquids are flavored with menthol/mint. There are 2433 products that contain alcohol flavors, while 3873 e-liquid products contain flavors of beverages other than alcohol and coffee/tea. In our sample, 1244 e-liquids contain tobacco flavors. The three main categories, nuts, spices/pepper, and coffee/tea are seen relatively less in our sample of e-liquids sold in online stores.

In Figure 2, we present a frequency plot of the 14,477 e-liquids in our sample, by classifying their flavor description into one of the following: (1) unflavored or flavor unknown, i.e., the product either is flavorless, or contains flavor(s) that are not identified by our key terms; (2) fruity flavor(s) only, i.e., the product contains one or more fruity flavors, and no additional flavor from any of the other main categories; (3) dessert/candy/other sweet flavor(s) only; (4) tobacco flavor only; (5) menthol/mint flavor only; (6) the product contains flavor(s) from one of the other flavor main categories, i.e., nuts, spices/pepper, coffee/tea, alcohol, or other beverages; (7) a combination of fruity and dessert/candy/other sweet flavors; (8) fruity and menthol/mint flavors; (9) a combination of fruity flavor(s), and beverage(s) other than alcohol and coffee/tea; (10) a combination of dessert/candy/other sweets, and tobacco; (11) dessert/candy/other sweets, and beverage(s) other than alcohol and coffee/tea; (12) containing flavors from two different main categories, but other than the ones described above; (13) a combination of fruit, menthol/mint, and dessert/candy/other sweets; (14) a combination of fruit, alcohol, and dessert/candy/other sweets; (15) a combination of fruit(s), dessert/candy/other sweets, and beverage(s) other than alcohol and coffee/tea; (16) a combination of fruit, menthol/mint, and beverage(s) other than alcohol and coffee/tea; (17) containing flavors from three different main categories, but other than the ones described above; (18) containing flavors from four or more different main categories. Among them, (2)–(6) describe single-flavored e-liquid products, i.e., containing flavor(s) from one main category only; (7)–(12) indicates a mix of flavors from two main categories; (13)–(17) mean the products each contains a mix of flavors from three main categories; and (18) describes e-liquid products in our sample that each contains a mix of flavors from four or more different main categories. One thing worth noting is that e-liquids with tobacco flavor only (total N = 230) may contain nicotine (N = 193; 83.91%) or be nicotine-free (N = 37; 16.09%).

As shown in Figure 2, the most prevalent flavor based on the count of products, is a combination of fruity and dessert/candy/other sweet flavors (N = 2872 products). The single-flavored e-liquids with fruity flavor(s) are also quite prevalent (N = 2472 products), followed by flavor description of four or more flavor main categories in e-liquid (N = 1787 products). A total of 1095 e-liquid products contain a combination of fruity and menthol/mint flavors; and there are 844 products with combined flavors of fruit(s), dessert/candy/other sweets, and beverage(s) other than alcohol and coffee/tea; in our sample, among 795 e-liquid products, each of them contains a combination of fruity flavor(s), and beverage(s) other than alcohol and coffee/tea. There are 417 products containing a combination of fruit, menthol/mint, and dessert/candy/other sweet flavors, while 496 e-liquids are flavored with a combination of fruit, alcohol, and dessert/candy/other sweets. A mix of two or more flavors from different main categories seems very common in e-liquids sold by online vape shops. We plot in Figure 3 the top seven flavors by frequency counts in each online store. The top e-liquid flavors by frequency counts in store 1 (from highest to lowest) are: fruity flavor(s) only (36.4%); fruity and dessert/candy/other sweet flavors (17.6%); fruity and menthol/mint flavors (7.6%); two different main categories, other than (7)–(11) (6.1%); three different main categories, other than (13)–(16) (5.6%); fruity flavor(s), and beverage(s) other than alcohol and coffee/tea (5%); and tobacco flavor only (3.8%). The top seven e-liquid flavors in store 2 are mostly similar to those in store 1: fruity flavor(s) only (21.5%); fruity and dessert/candy/other sweet flavors (20.3%); fruity and menthol/mint flavors (13.6%); fruity flavor(s), and beverage(s) other than alcohol and coffee/tea (8.6%); three different main categories, other than (13)–(16) (7.4%); two different main categories, other than (7)–(11) (6.1%); and a combination of flavors from four or more different main categories (4.3%). By frequency counts, the top seven (from highest to lowest) in store 3 are the following: fruity and dessert/candy/other sweet flavors (22.9%); fruity flavor(s) only (21.3%); two different main categories, other than (7)–(11) (7.7%); fruity and menthol/mint flavors (6.8%); unflavored or flavor unknown (6.3%); three different main categories, other than (13)–(16) (6.3%); fruity flavor(s), and beverage(s) other than alcohol and coffee/tea (6.1%). A combination of flavors from four or more different main categories is the most prevalent flavor by frequency counts, in both store 4 (27.9%) and store 5 (27.4%). It is likely that in both stores, e-liquid product description box contains links that direct customers to different flavored products in the same collection, and thus more than three different main categories were captured during keyword matching. In store 4, what follows are: fruity and dessert/candy/other sweet flavors (17.7%); containing flavors from three different main categories, other than (13)–(16) (10%); a combination of fruit(s), dessert/candy/other sweets, and beverage(s) other than alcohol and coffee/tea (8.3%); fruity flavor(s) only (6.4%); fruity and menthol/mint flavors (5.3%); a combination of fruity, menthol/mint, and dessert/candy/other sweet flavors (4.5%). Among e-liquids in store 5, the 2nd to 7th highest are: fruity and dessert/candy/other sweet flavors (17.8%); a combination of fruit(s), dessert/candy/other sweets, and beverage(s) other than alcohol and coffee/tea (12%); a combination of fruit(s), dessert/candy/other sweets, and alcoholic beverage(s) (8.9%); containing flavors from three different main categories, other than (13)–(16) (8.4%); a combination of fruity, menthol/mint, and dessert/candy/other sweet flavors (6.8%); fruity and menthol/mint flavors (4.9%).

## 4. Discussion

To understand and keep up with a variety of emerging e-liquid flavors available in the market of e-cigarettes, innovations in data sciences such as keyword matching has been increasingly used for identifying and classifying e-cigarette flavors and other characteristics when conducting surveillance of e-cigarette brand websites [33], social media [33,34,35,36], and manufacturer scan [37]. However, the semantic databases for conducting such as identification and classifications are often not publicly accessible and may be outdated by the evolvement of the marketplace. To address this gap, this article publishes the keywords for identifying and classifying flavors that can be readily used by the research community to code and update flavors.

Another contribution of the semantic database is to allow for a more accurate and comprehensive flavor classification. For example, the e-cigarette flavors reported by Nielsen Retail Scanner Data are limited to products sold in traditional brick-and-mortar stores [38,39,40,41,42,43], and the flavors of these products only reflect the primary flavor on a package, which may not capture all the flavor labels that are given to a product in the marketplace. Our research study is, to the best of our knowledge, the first to categorize multiple flavor labels based on real-world marketing practice.

Our study is informative and timely in that, it not only provides a database to help tobacco control researchers classify and analyze e-liquid flavors, but also shows that e-liquid products sold online are marked with multiple and complex flavor profiles, such as fruit + dessert, fruit + menthol, and fruit + spice, etc. Given that these attractive flavors are likely the most important attribute motivating adolescents and young adults to experiment with e-cigarettes and initiate vaping, restricting the flavor availability from online vendors could be important for curbing e-cigarette use among young populations [44,45,46,47,48,49,50]. Interestingly, e-liquids flavored with tobacco only do not make to the top seven by frequency count in any store except store 1, which has policy implication relevant to the role of tobacco-flavored e-cigarettes in smoking gateway pathway [16,20,21,22].

Flavor restrictions have been increasingly adopted by federal, state, and local policies to curb e-cigarette use. In recent years, the FDA issued Marketing Denial Orders (MDOs) to approximately 1 million flavored e-cigarettes and banned characterizing flavors other than menthol/mint and tobacco in cartridge-based e-cigarettes [27]. However, whether online vendors comply with these federal regulations are unknown [48,51,52]. Moreover, a growing number of states and localities, including New Jersey, New York, and San Francisco, California have adopted local restrictions to reduce the availability of flavored e-cigarette products [53]. These local flavor restrictions may incentivize e-cigarette users of flavored products to purchase products online to avoid regulations. More research is needed to ascertain the scale of online supply and purchases of flavored products and whether these activities undermine the impact of flavor restrictions on curbing e-cigarette use among young people.

Monitoring e-cigarette flavor profiles is also critical for understanding the relative appeals of nicotine and tobacco products in the marketplace. In April 2022, the FDA has issued two proposals that would ban menthol as a characterizing flavor in cigarettes and all characterizing flavors other than tobacco in cigars—an action to reduce the appeal of combustible tobacco [54,55,56]. If these policies are implemented, the availability of menthol and other attractive flavors in e-cigarettes could incentivize combustible tobacco smokers to transition into e-cigarettes. The cost and benefit of regulating e-cigarette flavors therefore need to be carefully weighted.

Finally, with a semantic database developed for flavors, our study could shed light for future research that identifies “concept flavors”—a group of ambiguous or vague descriptors that evoke sensory or other pleasant experiences (e.g., jazz, ice, etc.). Concept flavors could replace characterizing flavors as a marketing strategy and continue to attract consumers when products with explicit flavors are banned by authorities [43,57,58]. Our database will allow researchers to identify products that do not fall into any of the known flavor categories, thereby assisting in the evaluation of whether these producers adopt concept flavors to market these products.

Our study has several limitations. The flavor semantic database contains pre-determined hierarchies and categories of flavors based on the current consensus, which could change over time. Nonetheless, users of our database may modify hierarchies and categories as needed to reflect the development of marketplace. In addition, although we used several iterations of keyword matching and expert reviews to enrich the flavor vocabulary, the database is not an exhaustive list of all flavor-related words. We welcome readers and users of this paper to improve and augment the database and make their contributions publicly accessible to facilitate future research. Finally, although our approach could assist in identifying concept flavors, such effort is beyond the scope of this paper. We will conduct a follow-up study to specifically investigate concept flavors.

## 5. Conclusions

Flavor is an important attribute of e-liquid products, especially to youth users of e-cigarettes. Development of new e-liquid flavors could be associated with initiation and escalation of e-cigarette use among U.S. adolescents and adults. Regulations on e-liquid flavors are under the authority of the U.S. Food Drug Administration. The findings of this study can be used not only to inform potential policies, but also to shed light on future research directions regarding e-cigarette flavors.

## Figures and Tables

**Figure 1 ijerph-19-13953-f001:**
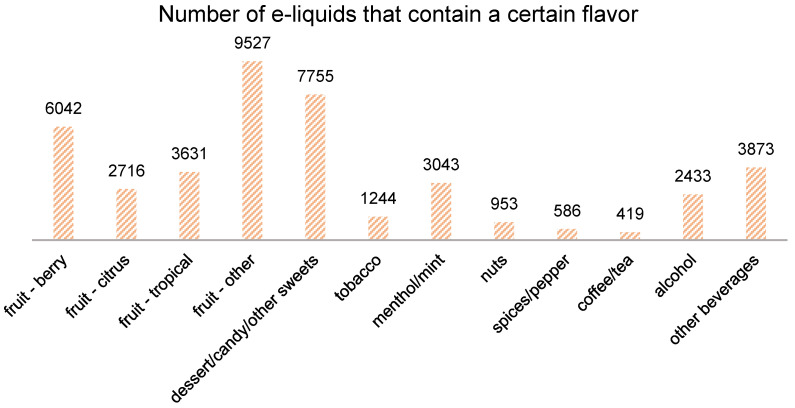
Frequencies of E-liquids Containing a Certain Flavor (Product Total N = 14,477).

**Figure 2 ijerph-19-13953-f002:**
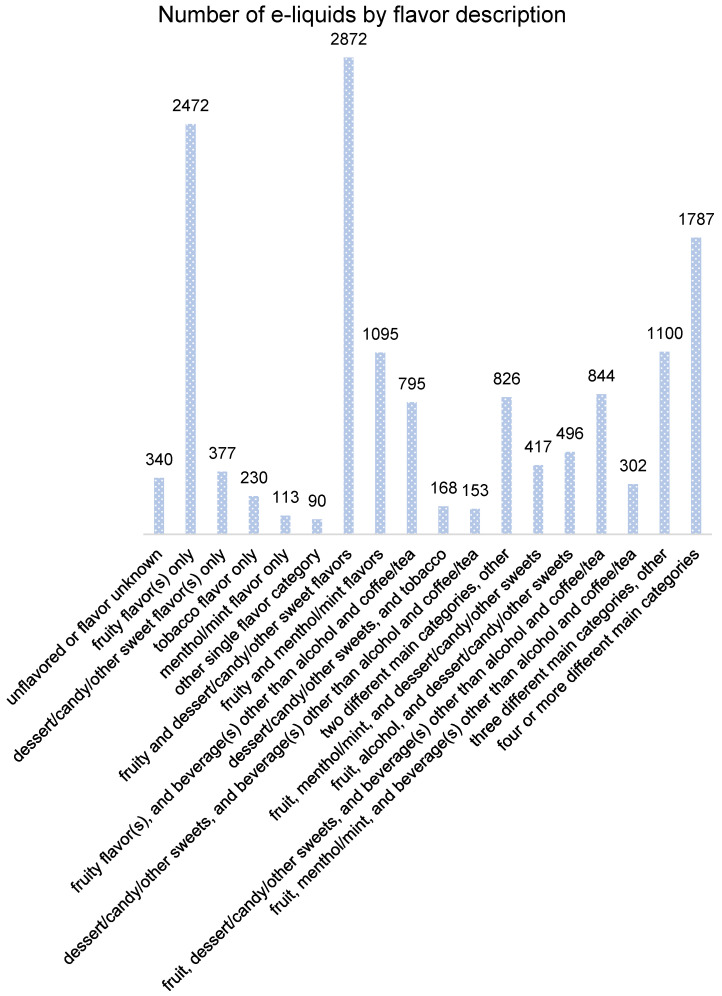
Frequencies of E-liquids Based on Flavor Description (Product Total N = 14,477).

**Figure 3 ijerph-19-13953-f003:**
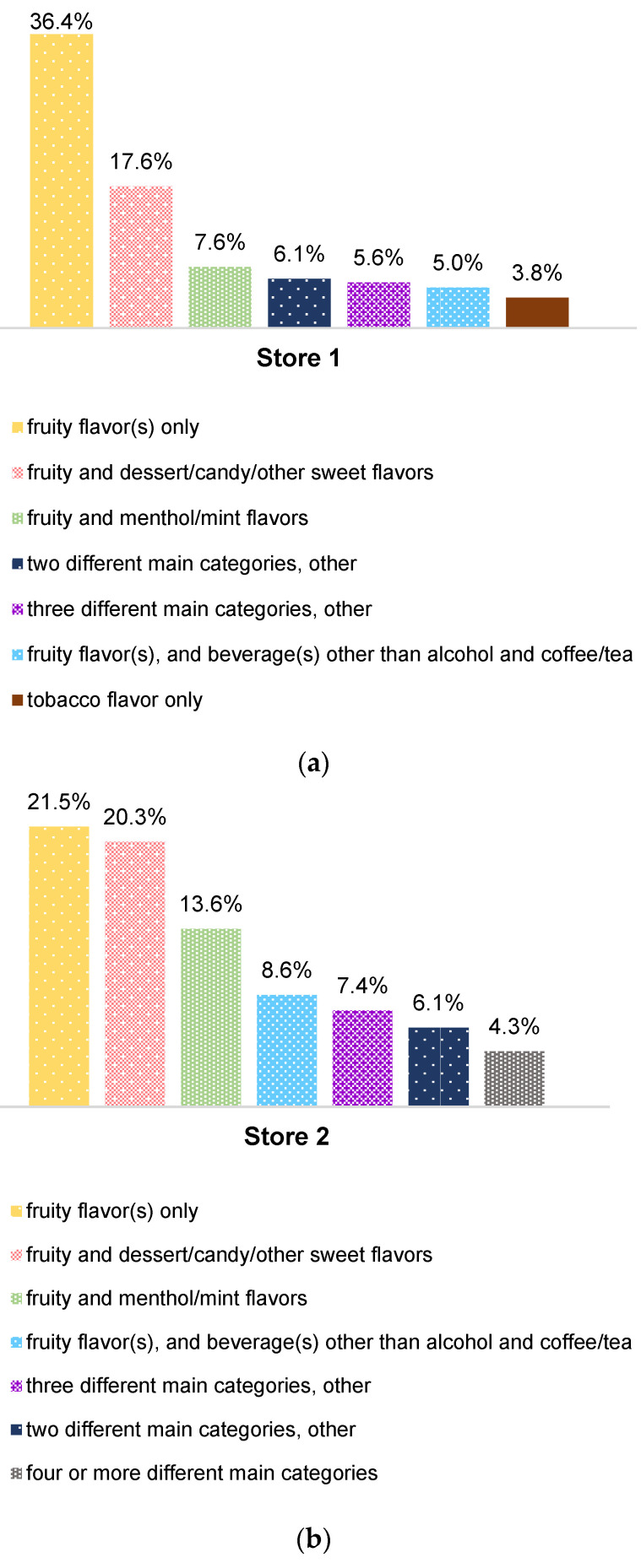
Top Seven Flavors in Each Online Vape Shop by Frequency Counts (Subfigures (**a**–**e**) show the top seven flavors by frequency counts in stores 1–5 respectively).

## Data Availability

The price data was made available through another open-access publication (DOI: 10.1136/tobaccocontrol-2021-057033). We will also publish the data with this study.

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
