# Peer review of "Expanding the E-Liquid Flavor Wheel: Classification of Emerging E-Liquid Flavors in Online Vape Shops"

_ijerph, 2022, doi:10.3390/ijerph192113953_

Round 1

Reviewer 1 Report

The rapid expansion of e-cigarette use among young people represents, indeed, a serious public health issue. The fact that the addition of flavors to e-liquids is associated with an increased consumption tendency, especially among young people, is sustained by several research articles.

In the context of the restrictions imposed on commercialization of flavored e-cigarettes, this article is useful for flavors' classification and identification. 

The research was meticulous, and the number of screened e-liquids was very high. 

The authors covered the aspects I can think of related to the subject, so I only have some minor observations/suggestions:

  • the term "concept flavors" appears first in the Introduction section, but its explanation is given only in the Discussion section; for improved clarity I would insert a short explanation in the Introduction section as well
  • Line 100 - Ma et al (2022) - reference needed
  • Line 312 - WordNet - wrong reference number.

Author Response

  1. Comments to the Author

The rapid expansion of e-cigarette use among young people represents, indeed, a serious public health issue.

The fact that the addition of flavors to e-liquids is associated with an increased consumption tendency, especially among young people, is sustained by several research articles.

In the context of the restrictions imposed on commercialization of flavored e-cigarettes, this article is useful for flavors' classification and identification.

The research was meticulous, and the number of screened e-liquids was very high.

The authors covered the aspects I can think of related to the subject, so I only have some minor observations/suggestions.

Response: We thank the reviewer for commenting on the contribution of our paper. We hope the expanded flavor wheel will serve as a helpful tool for tobacco control researchers to classify and identify the new and emerging e-liquid flavors in the marketplace of e-cigarettes.

  1. Comment: the term "concept flavors" appears first in the Introduction section, but its explanation is given only in the Discussion section; for improved clarity I would insert a short explanation in the Introduction section as well

Response: Thank you for this suggestion. We have revised our introduction section accordingly.

  1. Comment: Line 100 - Ma et al (2022) - reference needed.

Response: Thank you for this comment. We have added the reference.

  1. Comment: Line 312 - WordNet - wrong reference number.

Response: Thank you for pointing this out. We have corrected the reference number.

Reviewer 2 Report

Since I agreed, the review should be written. The doubts are explained by the fact that I am a chemist, but the sense of the article is problems of semantics, namely, building the semantic database for flavors.

Nevertheless, the content of the article and Authors’ position seems intriguing and attractive. Firstly, it concerns the positioning of the problem under consideration itself. It is highly likely that a similar semantic viewing could be useful in other areas. Moreover, there is an important Authors’ statement, which is located at the page 9, lines 247, 248: “… the availability of menthol and other attractive flavors in e-cigarettes could incentivize combustible tobacco smokers to transition into e-cigarettes”. The reviewer proposes to repeat this phrase (or something similar) in the Abstract.

Some small corrections can be recommended for Authors. At first, the definition of e-cigarette (page 1, line 42): “… it is a device that atomizes a liquid solution …”. So far as the e-cigarette is not A-bomb, the word “atomizes” seems better to replace with “evaporates”.

Page 3, line 109: Figure A1 is mentioned. The first-time-reader of the text few seconds remains left at a loss as to where to find this Figure. Please indicate “Figure A1 (Appendix)”.

The main fragment of the Section 3 (Results) is located at page 5. Unfortunately, the text at this page looks like large monolith block with 198 – 155 = 43 lines without its subdivision onto any paragraphs, which is somewhat uncomfortable for reading. Besides that, this fragment contains a lot of numerical data inside it. Thus, for optimizing the structure of the article, the Authors should consider the possibility of additional subdividing of this text.

Mentioned above is equivalent to the minor revision of the text before its publication.

Author Response

  1. Comments to the Author

Since I agreed, the review should be written. The doubts are explained by the fact that I am a chemist, but the sense of the article is problems of semantics, namely, building the semantic database for flavors.

Nevertheless, the content of the article and Authors’ position seems intriguing and attractive. Firstly, it concerns the positioning of the problem under consideration itself. It is highly likely that a similar semantic viewing could be useful in other areas. Moreover, there is an important Authors’ statement, which is located at the page 9, lines 247, 248: “… the availability of menthol and other attractive flavors in e-cigarettes could incentivize combustible tobacco smokers to transition into e-cigarettes”. The reviewer proposes to repeat this phrase (or something similar) in the Abstract.

Response: Thank you for this comment. We have revised the abstract accordingly.

  1. Comment: Some small corrections can be recommended for Authors. At first, the definition of e-cigarette (page 1, line 42): “… it is a device that atomizes a liquid solution …”. So far as the e-cigarette is not A-bomb, the word “atomizes” seems better to replace with “evaporates”.

Response: Thank you for this comment. We have replaced “atomizes” with “evaporates”.

  1. Comment: Page 3, line 109: Figure A1 is mentioned. The first-time-reader of the text few seconds remains left at a loss as to where to find this Figure. Please indicate “Figure A1 (Appendix)”.

Response: Thank you for this suggestion. We have revised it accordingly.

  1. Comment: The main fragment of the Section 3 (Results) is located at page 5. Unfortunately, the text at this page looks like large monolith block with 198 – 155 = 43 lines without its subdivision onto any paragraphs, which is somewhat uncomfortable for reading. Besides that, this fragment contains a lot of numerical data inside it. Thus, for optimizing the structure of the article, the Authors should consider the possibility of additional subdividing of this text.

Mentioned above is equivalent to the minor revision of the text before its publication.

Response: Thank you for this comment. We have made the change accordingly.

Reviewer 3 Report

This is an interesting and relevant study that addressed the research gaps by identifying and categorising e-liquid flavours of over 14,000 products sold online. The analyses seem to have been executed correctly and the article is well-written.

Overall, the introduction gives a very clear description of the evidence to date and the gap in evidence that this study fills. It was mentioned in the manuscript that the development of new e-liquid flavors could be associated with the initiation and escalation of e-cigarettes (Introduction: Paragraph 1) and e-cigarettes could serve as a gateway to cigarette smoking (Introduction: paragraph 3). However, rationalization of the use of different flavours (such as fruit, mint/menthol, dessert/candy/sweets) mixed with nicotine-containing e-cigarettes and their impact on the smoking gateway pathway is worth mentioning.

Nicotine is a known addictive substance in tobacco. Pathophysiological changes induced by continued nicotine exposure among tobacco users result in addiction. In this manuscript, figure 2 shows, the frequency plot of the e-liquids in the sample, where the 4th category shows tobacco flavour (N=230) only, it will be helpful if the author can clarify the meaning of tobacco flavour. Are those nicotine-containing flavour? Similarly, Figure 3 shows the top seven flavors by frequency counts of online store 1 to store 5, only store 1 shows tobacco flavour (3.8%) which is worth mentioning in the discussion section as e-cigarettes containing tobacco flavour have a contribution to the smoking gateway pathway.  

Author Response

  1. Comments to the Author

This is an interesting and relevant study that addressed the research gaps by identifying and categorising e-liquid flavours of over 14,000 products sold online. The analyses seem to have been executed correctly and the article is well-written.

Response: We thank the reviewer for commenting on the research gaps that our paper addresses.

  1. Comment:

Overall, the introduction gives a very clear description of the evidence to date and the gap in evidence that this study fills. It was mentioned in the manuscript that the development of new e-liquid flavors could be associated with the initiation and escalation of e-cigarettes (Introduction: Paragraph 1) and e-cigarettes could serve as a gateway to cigarette smoking (Introduction: paragraph 3). However, rationalization of the use of different flavours (such as fruit, mint/menthol, dessert/candy/sweets) mixed with nicotine-containing e-cigarettes and their impact on the smoking gateway pathway is worth mentioning.

Response: Thank you for the insightful comments. We have revised the introduction section accordingly.

  1. Comment:

Nicotine is a known addictive substance in tobacco. Pathophysiological changes induced by continued nicotine exposure among tobacco users result in addiction. In this manuscript, figure 2 shows, the frequency plot of the e-liquids in the sample, where the 4th category shows tobacco flavour (N=230) only, it will be helpful if the author can clarify the meaning of tobacco flavour. Are those nicotine-containing flavour?

Response: Thank you for this suggestion. The information that an e-liquid product contains tobacco flavor is from the source code of the product webpage, or flavor filter on the store website, or product description text on the product webpage, i.e., that product is labeled as tobacco-flavored by the store website. We have added “One thing worth noting is that e-liquids with tobacco flavor only (total N = 230) may contain nicotine (N = 193; 83.91%) or be nicotine-free (N = 37; 16.09%).”

  1. Comment:

Similarly, Figure 3 shows the top seven flavors by frequency counts of online store 1 to store 5, only store 1 shows tobacco flavour (3.8%) which is worth mentioning in the discussion section as e-cigarettes containing tobacco flavour have a contribution to the smoking gateway pathway.

Response: Thank you for this comment. We have added the following sentence to the discussion section: “Interestingly, e-liquids flavored with tobacco only do not make to the top seven by frequency count in any store except store 1, which has policy implication relevant to the role of tobacco-flavored e-cigarettes in smoking gateway pathway.[16,20–22]”.